# Tongue Rehabilitation Device for Dysphagic Patients

**DOI:** 10.3390/s19214657

**Published:** 2019-10-26

**Authors:** Mario Milazzo, Andrea Panepinto, Angelo Maria Sabatini, Serena Danti

**Affiliations:** 1The BioRobotics Institute, Scuola Superiore Sant’Anna, Viale Rinaldo Piaggio 34, 56025 Pontedera (PI), Italy; angelo.sabatini@santannapisa.it; 2Department of Surgical, Medical, Molecular Pathology and Emergency Care, University of Pisa, via Savi 10, 56126 Pisa, Italy; a.panepinto@studenti.unipi.it; 3Department of Civil and Industrial Engineering, University of Pisa, Largo L. Lazzarino, 2, 56126 Pisa, Italy

**Keywords:** rehabilitation, dysphagia, tongue, personalized medicine

## Abstract

Dysphagia refers to difficulty in swallowing often associated with syndromic disorders. In dysphagic patients’ rehabilitation, tongue motility is usually treated and monitored via simple exercises, in which the tongue is pushed against a depressor held by the speech therapist in different directions. In this study, we developed and tested a simple pressure/force sensor device, named “Tonic Tongue (ToTo)”, intended to support training and monitoring tasks for the rehabilitation of tongue musculature. It consists of a metallic frame holding a ball bearing support equipped with a sterile disposable depressor, whose angular displacements are counterbalanced by extensional springs. The conversion from angular displacement to force is managed using a simple mechanical model of ToTo operation. Since the force exerted by the tongue in various directions can be estimated, quantitative assessment of the outcome of a given training program is possible. A first prototype of ToTo was tested on 26 healthy adults, who were trained for one month. After the treatment, we observed a statistically significant improvement with a force up to 2.2 N (median value) in all tested directions of pushing, except in the downward direction, in which the improvement was slightly higher than 5 N (median value). ToTo promises to be an innovative and reliable device that can be used for the rehabilitation of dysphagic patients. Moreover, since it is a self-standing device, it could be used as a point-of-care solution for in-home rehabilitation management of dysphasia.

## 1. Introduction

Swallowing is defined as the voluntary physiologic process that transports ingested material and saliva from the mouth to the stomach. However, between meals, it occurs also without any conscious input [1]. The tongue is a key organ enabling the swallowing process. It moves food around the mouth within the oral cavity by pressing it against the hard palate and out to the sides to allow mastication. The food bolus is thus formed in the oral preparatory phase of swallowing [2]. The tongue also takes part in the pharyngeal phase of swallowing by elevating and sweeping posteriorly to propel the food bolus past the anterior tonsillar pillar, triggering the swallowing reflex [3]. Dysphagia is defined as when any difficulty in swallowing occurs, usually as a consequence of neural, such as stroke, and neurodegenerative pathologies affecting the muscles involved in deglutition, among others. Dysphagia is a threatening pathology in the elderly, as non-swallowed food debris can easily be transported to the trachea and therefore into the lungs by breathing, which are a major cause of pneumonia [4]. Normal swallowing consists of four phases: oral preparatory, oral, pharyngeal, and esophageal. Stroke usually affects the first three phases by interrupting the voluntary control of chewing and moving food around the mouth or delaying the pharyngeal reflex [5].

Rehabilitation strategies for dysphagic patients consist of compensatory, direct, and indirect methods. The compensatory approach is used as a temporary technique in view of more specific approaches. Direct strategies are thus employed to quicken the recovery process by promoting a modification of the swallowing physiology, through sensorial stimulations and motor redirection of the bolus. Finally, indirect techniques include stimulation of oral and pharyngeal structures without any specific implication to the swallowing task; they aim at strengthening the swallowing musculature and improving its motility [6]. Patients with dysphagia, often caused by strokes or apraxia, show a reduced capability to propel bolus into the pharynx through tongue musculature [7,8,9].

The traditional approach to tongue rehabilitation in dysphagic patients consists of isometric and isotonic exercises. An isometric lingual exercise regimen results in significantly increased isometric and swallowing pressures, as well as in changes in tongue volume measured with anatomic magnetic resonance imaging [10]. Robbins et al., for instance, examined effects of lingual exercises on various aspects of swallowing recovery post-stroke including lingual anatomy [11].

In contrast, isotonic exercises feature repeated sessions of tasks in which a speech therapist firmly holds a disposable depressor against the patient’s tongue. Based on the direction of the force exerted against the tongue, the exercises are classified as: forward, lateral (right/left), and vertical (up/down) thrusts Figure 1.

Although largely used, this approach presents drawbacks, namely, (i) discontinuous training, since a speech therapist is required for a correct performance of the exercises; (ii) non-consistent training, since the speech therapist applies an unknown counterforce; and (iii) subjective evaluation regarding patient performance and recovery, which are all based on the perception of the speech therapist.

In order to monitor subjects’ progress, devices have been developed to provide some measurement of the tongue pressure after only isometric tasks; moreover, they have been conceived, almost exclusively, for research purposes and have not considered invasiveness to any relevant extent. A first case is a small pressure sensor to be placed on the palatal appliance, which turns out to be both non-sterile and expensive [12,13]. Robbins et al. designed a balloon-based system for clinical applications but due to the difficulties in classifying standard values for tongue pressures, its employment has not been largely pursued [2,14]. Recently, Utanohara et al. developed a disposable tongue pressure measurement device to evaluate deviations in different cohorts of patients; in particular, they showed that age affects tongue muscular performance, with men showing a faster strength reduction rate than women [15,16,17]. Hori et al. proposed a sensorized layer to be implanted on the palate, to measure tongue pressure during swallowing. Although interesting, this is an invasive method that dysphagic patients can find difficult to accept [18]. Finally, Abilex* (TMI Medical Distribution Inc., London, ON, Canada) is a commercial device that has been used to improve the intraoral pressure against the oral walls through isometric tasks. However, Abilex* does not provide a tunable counterforce, as it is a standard tool that cannot be adapted to patients’ specific needs [19].

To the best of our knowledge, the “Tonic Tongue” (ToTo) device presented in this paper is the only non-invasive equipment that, in contrast to state-of-the-art approaches, simultaneously provides assistance to the performance of isotonic exercises for tongue strength rehabilitation and direct, reliable monitoring through force measurements. ToTo was, indeed, conceived to perform traditional isotonic exercises as proposed by speech therapists in a consistent way, with a reliable quantification of tongue force. The simple design of the structure, associated with the possibility to tune the treatment, in principle, makes ToTo a unique point-of-care solution for at-home rehabilitation.

In this paper, we describe the main features of the ToTo system, and we present preliminary results of a pilot study, to assess ToTo feasibility and effectiveness with healthy subjects, in preparation for a future experimental study planned with dysphagic patients.

## 2. Materials and Methods

### 2.1. ToTo Architecture and Working Principle

ToTo is a portable device designed to rehabilitate people that are affected by dysphagia through simple exercises, aimed to improve the muscular tone of the tongue. Its dimensions are 145 × 87 × 128 mm (Figure 2). The main frame of ToTo is composed of two specular steel parts, in each of which a ball bearing is forced through an external ring; a roller is fixed into the internal ring, free to rotate about the central axis; a sterile disposable tongue depressor can be inserted in the central axis. A graduated angle scale, with resolution 5° that corresponds to a force of about 2.2 N in the current implementation of the ToTo prototype (see Table 1), is positioned on the exterior of the frame; two traction springs are placed on both sides of the frame between the rollers and the frame itself, so as to equilibrate the torque generated by the tongue. Based on the subject’s needs, it is possible to modify the passive resistance exerted by the traction springs by changing their elastic constant or the spring number.

Figure 3 shows a simplified model of ToTo mechanical behavior based on the use of a single traction spring. When a subject pushes the tongue depressor, a pressure p is exerted over a finite portion of the depressor. For simplicity in the model development, the overall force $F_{L}$ is assumed to be concentrated and the depressor is a rigid part; the resting length $L_{0}$ of the spring is then changed to its final value L, according to the equation:(1)$$L=\sqrt{{\left(L_{0}+R\left(1-\cos\alpha \right)\right)}^{2}+{\left(R\sin\alpha \right)}^{2}},$$
where α is the rotation angle of the bearing, measured through the graduated scale, and R is the radius of the roller where the spring is connected.

The restoring force *F_m_* exerted by each spring can be written as:(2)$$F_{m}=F_{0}+k\left(L-L_{0}\right)$$
where $F_{0}$ is the preload and k is the spring elastic constant. 

Based on simple geometrical considerations, the following equation specifies the relationship existing between $F_{L}$ and $F_{m}$ in the case of n springs:(3)$$F_{L}=F_{m}\cos\delta \frac{nR}{d\cos\alpha }\;,\;\mathrm{with}\;\delta =\frac{\pi }{2}-[\alpha +\sin^{-1}(\frac{R\sin\alpha }{L})]$$
where d is the distance from the center of the bearings to the point of application of the concentrated force by the tongue. Numerical values of the model parameters for the actual implementation of ToTo are reported in Table 1. Data collected as values of angular displacement can be converted into values of force using Equation (3).

### 2.2. Experimental Protocol

The sample of subjects enrolled consisted of 26 healthy adults (15 male and 11 female, with 18 subjects under 35 years) without dysphagic problems who were recruited by a speech therapist (author A.P.). All the subjects declared that they were not taking medication at the time of the experiments. After being informed about ToTo and the procedures involved in the experimental design, they provided written consent to participate in the training program. 

Although it is possible to change the number of springs and their stiffness according to the specific patient need, we used the configuration of ToTo summarized in Table 1 for all the volunteers since nobody had any dysphagic issue. Ultimately, this approach ensured the consistency of the results that were submitted to statistical analysis. 

Participants were asked to perform five different isotonic exercises of 5 min each, avoiding any head/neck movement, under the direct supervision of the speech therapist. In each exercise tongue thrusts were generated in a specific direction, namely forward (exercise #1); lateral—right and left (exercise #2 and exercise #3); and vertical—up and down (exercise #4 and exercise #5); as shown in Figure 4. In each exercise, after allowing a few minutes of familiarization with the task, the duty cycle consisted of two repetitions of 20 s of active work with 10 s intervals. We did not counterbalance the order of presentation of exercises to participants, since the final outcome of the training program was the global reinforcement of the tongue musculature.

Each participant performed 10 sessions during a 1-month timeline. Immediately before the start of the training program (time t_0_) and immediately after completion of the training program (time t_1_), subjects were asked to apply the highest force they were capable of producing in each direction. As a further score of performance, the omni-directional force, defined as the average of the five values of maximal forward, lateral and vertical forces, was also computed and submitted to statistical analysis.

### 2.3. Statistical Analysis

The sign test was used to determine whether there was a median difference between the maximal force applied by participants (forward, right, left, up, down, omni-directional) at the two time points, *t*_0_ and *t*_1_. The sign test is considered an alternative to the paired-samples *t*-test or Wilcoxon signed-rank test, since the distributions of differences turned out to be neither normal nor symmetric in all directions. The statistical analysis was performed using the IBM SPSS Statistics software package (IMB SPSS Statistics 26, SPSS IBM, New York, NY, USA).

## 3. Results

Of the 26 participants recruited to the study, the training program elicited an increase in the maximal force applied in forward, right, left and up directions in 24 participants, whereas two participants performed the same (i.e., two ties were observed); in the downward direction, improvements were found in 25 participants, with only one tie. As for the omni-directional force, improvements were detected in all participants with no ties. The subjects involved in the ties were never the same for each loading direction; moreover, no participant performed worse with the training program, regardless of the experimental condition.

Histograms of the differences of related samples, $∆F=F\left(t_{1}\right)-F\left(t_{0}\right)$, for each loading direction and in the omni-directional case are reported in Figure 5.

There was a statistically significant median increase in maximal force applied in the forward direction (2.57 N) at *t*_1_ (7.65 N) compared with that at *t*_0_ (4.03 N), *z* = 4.695, exact *p* < 0.0005. The term *z* denotes the standardized test statistic; the exact *p*-value (two-sided test) was computed based on the binomial distribution because there were 25 or fewer cases; in the case of the omni-directional force, the asymptotic *p*-value (two-sided test) was considered because the number of cases was 26. With this interpretation of the relevant statistical quantities, please refer to Table 2 for a report of the results in each direction and for the omni-directional force. Finally, note that we did not observe any significant effect on the results due to the low elastic or permanent deformation of the tongue depressor, also because of the very small loads involved and the limited resolution of the instrument. This confirmed our hypotheses and the reliability of the model.

## 4. Discussion and Conclusions

We have presented a novel pressure/force sensor device, named ToTo, which holds promise for advancing the rehabilitation of tongue musculature in dysphagic patients. It consists of a solid frame with rotary supports for a sterile disposable tongue depressor, which is able to counteract the thrusts exerted by patient’s tongue through traction springs. ToTo, as a simple, self-standing and transportable system, can serve as a point-of-care system to improve and monitor the tongue musculature performance through isotonic extra-oral exercises. In this study, we explained ToTo’s architecture and mechanics and assessed its performance based on the results of a pilot study involving 26 healthy subjects without any the control group, according to the guidelines of the local Ethical Committee for a proof of concept of the device.

Our results showed a statistically significant difference in tongue force measured before and after the treatment, with an average increase of ~3 N. By analyzing the results obtained by pressing the tongue in different directions, we observed the highest improvement for exercise #5–downward thrust (~5 N), while lateral pushing exercises show the smallest increase (~2 N). As for exercise #4 (upward push) only, we are in the position to compare our results with those provided by previous research reports, particularly those in which the authors measured tongue pressures on the palate through the interposition of inflatable balloons or sensorized polymeric layers [17,20]. Such measured pressures were reported in the order of 40–60 kPa, which, based on geometrical considerations, may correspond to ~0.7–1 N [17,20]. These are lower values, when compared with the ones we obtained in this study. We believe that the discrepancy is due to the different measuring approaches adopted: we collected data from ToTo using the mechanics of elastic bodies triggered by the tip of the tongue, modeled as a concentrated force, while earlier studies employed pneumatic/electronic systems that measure heterogeneous pressure distributions on the palate. While we employed a tongue positioning device that exploits all the musculature of the tongue stretched outside the mouth, previous studies performed their experimental protocols with the tongue contracted in the mouth without any specific requirement, or repeatability, in terms of muscle employment and positioning.

ToTo is a non-invasive system that augments the traditional isotonic exercises with non-null displacements, in contrast to the traditional isometric lingual strength-training (IOPI) [10]. Moreover, in contrast to the devices so far conceived for the IOPI, it does not require the positioning of foreign bodies into the oral cavity in the form of either a polymeric balloon (e.g., Abilex*) [12,13,14,17] or implantable sensor sheet [18]; moreover, in contrast to earlier studies, is not limited to being a mere measurement tool. ToTo, indeed, offers a unique double feature, working simultaneously as a rehabilitative and measuring device. The exercises that subjects can perform are similar to the traditional ones proposed by speech therapists, and ToTo use is not constrained to the continuous presence of a specialist who, however, can monitor and tune the treatment in the long term. This important feature would make ToTo a unique easy-to-use point-of-care tool for tongue rehabilitation, potentially synergistically coupled with other intra-oral devices (e.g., Abilex*), in patients without any specific neural disorder. In contrast, namely when apraxia occurs, patients are not able to intentionally use the tongue musculature, and ToTo may be used in combination with other approaches, such as the expiratory muscle strength training, usually employed in this scenario [21].

Although our results show an interesting and encouraging picture, a few margins of improvement are necessary and are part of our ongoing work. The first step will consider the redesign of the frame to embed miniaturized encoders for a more accurate measurement of the angular displacements, as well as the development of dedicated electronics to filter and store measured data. These data could be eventually sent to the speech therapist in real-time through a mobile app for further progress assessment. In its current form, ToTo is a fully mechanical device, which does not require any power supply. However, it does not record performance history, which would be necessary in the aspect of e-health monitoring.

Concerning system validation, our investigation provided only a preliminary assessment on healthy subjects without any dysphagic symptoms. Therefore, the ultimate challenge will be focused on diseased subjects, in order to discriminate how effective ToTo can be at different pathologic stages, for pursuing the swallowing task, and possibly for investigating the tongue behavior upon isometric pressures, potentially synergistically with other intra-oral devices. In this case, due to the difficulty in preventing unintentional force compensations with head/neck movements, we envisage the employment of a dedicated rigid frame, similar to the one used for eye-related treatments. Finally, ToTo will be assessed in terms of user experience (e.g., ease of use, user comfort).

## 5. Patents

Inventors: Andrea Panepinto, Stefania Santopadre, Serena Danti, Mario Milazzo. Title: Device for tongue muscle rehabilitation (“Dispositivo per la riabilitazione dei muscoli della lingua”). Patent submitted with Italian priority, by University of Pisa and Azienda Ospedaliero-Universitaria Pisana.

## Figures and Tables

**Figure 1 sensors-19-04657-f001:**
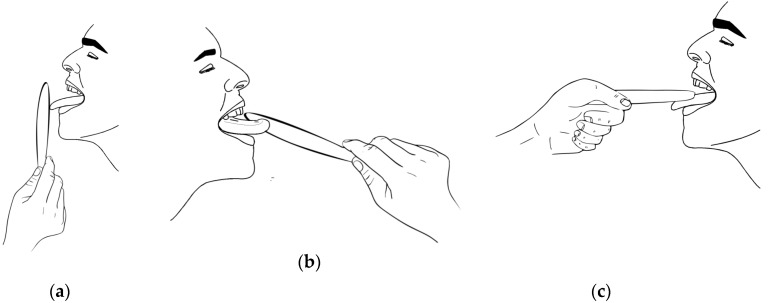
Traditional exercises proposed by speech therapists to treat and monitor tongue mobility rehabilitation. A speech therapist holds a disposable depressor against which dysphagic patients exert pressure with the tongue. Based on the directionality, we distinguish the exercises as (**a**) forward; (**b**) lateral (right/left); and (**c**) vertical (up/down) tasks.

**Figure 2 sensors-19-04657-f002:**
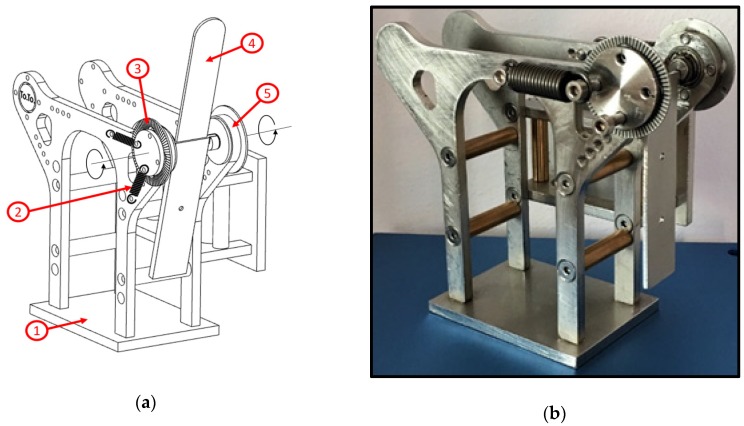
Architecture of ToTo device. (**a**) The axonometric view of ToTo presents the main components: (1) frame; (2) traction spring; (3) graduated scale; (4) disposable tongue depressor; (5) depressor holder with rollers supported by ball bearings. (**b**) A picture of the ToTo prototype implemented with two traction springs (one per side).

**Figure 3 sensors-19-04657-f003:**
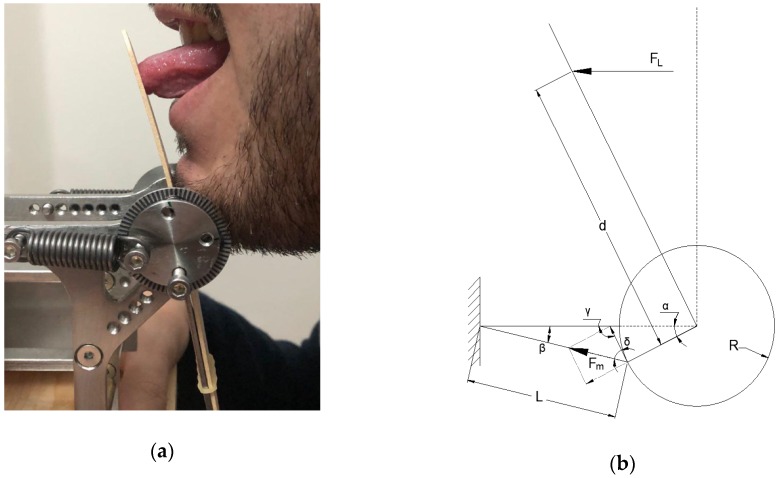
(**a**) Forward exercise performed by a volunteer. (**b**) Mechanical scheme of the forward thrust with depiction of the pushing force $F_{L}$ exerted by the tongue and the reaction force $F_{m}$ from the linear spring.

**Figure 4 sensors-19-04657-f004:**
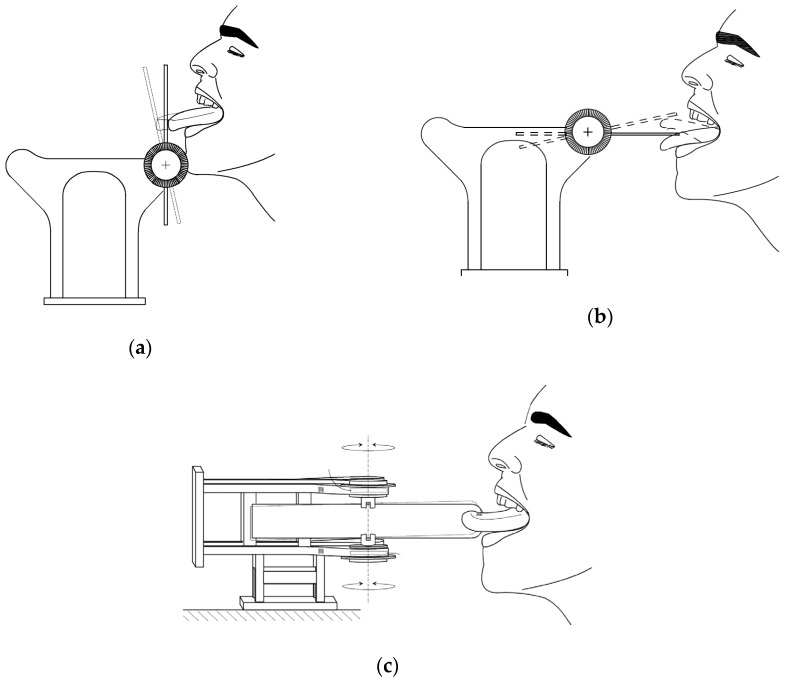
Experimental protocol to assess ToTo performance: (**a**) forward, (**b**) vertical (up and down), and (**c**) lateral (right and left) thrust exercises.

**Figure 5 sensors-19-04657-f005:**
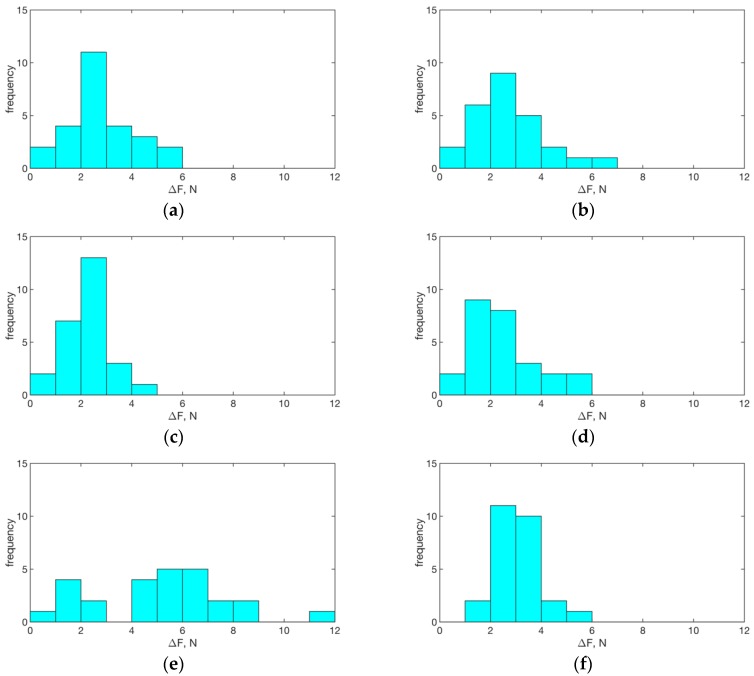
Histograms depicting the related-samples differences for each loading direction: (**a**) forward; (**b**) right; (**c**) left; (**d**) up; (**e**) down; and (**f**) omni-directional. We observe similarities among panels **a**–**d** and **f** with distributions having peaks between 2–3 N. Panel (**e**) presents, in contrast, a broader distribution of counts, with occurrences up to ~12 N, showing higher tongue musculature improvement.

**Table 1 sensors-19-04657-t001:** Numerical values of the parameters introduced in the mechanical model of Equation (3) for the current implementation.

Parameter	Value
d (mm)	35.0
R (mm)	10.0
$L_{0}$ (mm)	37.5
k (N·mm^−1^)	13.04
$F_{0}\;$ (N)	34.3
n	2

**Table 2 sensors-19-04657-t002:** Statistical reporting.

Direction	$F(t_{0})$	$F(t_{1})$	∆F	*z*	*p*-Value
Forward	4.03	7.65	2.57	4.695	<0.0005
Right	2.46	7.65	3.39	4.695	<0.0005
Left	2.46	4.67	2.21	4.695	<0.0005
Upward	2.21	4.67	2.38	4.695	<0.0005
Downward	6.08	11.40	5.32	4.800	<0.0005
Omni-directional	4.10	7.26	2.93	4.903	<0.0005

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
