# Peer review of "Tongue Rehabilitation Device for Dysphagic Patients"

_sensors, 2019, doi:10.3390/s19214657_

Round 1

Reviewer 1 Report

The given study investigates a newly developed pressure / force device to support rehabilitation of the tongue musculature, called “Tonic Tongue” (ToTo). The device will allow for training as well as measuring tongue strength in different directions of tongue movement. A prototype of the “ToTo” was used as training device in 26 healthy adults for a month and lead to a significant improvement of force in every direction but downward. Deriving from these data, it is hypothesized that the “ToTo” may be an appropriate tool for dysphagia rehabilitation in patients.

Dysphagia is a relevant implication of numerous diseases and may lead to aspiration (pneumonia), malnutrition, exsicossis, impaired quality of life and increased mortality. Only few devices have been developed so far to aid with swallowing rehabilitation, particularly devices that may be used outside of clinical care. Devices that can be used at home or in an ambulant rehabilitation setting may improve dysphagia care which – in my opinion – makes studies like the one given relevant. Furthermore, the device not only allows for training but also for measurement of training success. The language is mainly sound, figures are clear and make the functionality of “ToTo” easily understandable. The manuscript is well structured to the most part. In my opinion, a number of issues have to be addressed however:

Major concerns:

In the manuscript, I did not find a comment on any kind of approval of an ethics committee. This being an interventional study, such a vote is absolutely necessary and needs to be stated in the manuscript independently from written consent by the participants. The study design did not imply a control-group. To measure effects of “ToTo”, a control group without therapy as well as a control group performing today’s standard of care (logopedic training) (alternatively also using the IOPI for training as an alternative tongue strengthening measure, although mainly used only for the upward direction) would allow for a comparison. The training implied tasks on isometric performance for 20 seconds during 5 minutes for each direction, yet the results were reported on an increase of maximal force. Two aspects may be discussed with regard to this aspect: 1. Is there a reason why data of isometric performance were not part of the study protocol? 2. Maximal tongue force as well as higher isometric pressures seem to be relevant for swallowing efficiency during the oral phase of swallowing, however not being decided yet which of these parameters is more important. With regard to this study, no clear answer to this question can be given, yet this aspect would add to the discussion as “ToTo” in principle would allow to train either maximal force as well as isometric training.

Minor concerns:

In the introductory part, the tongue is entitled to be the “most important organ enabling the swallowing process”. In a complex sequence of more than 25 pairs of muscles that are involved in the act of swallowing it may be more suitable to call the muscle a major organ rather than the most important one. In line 38/39 the tongue is associated with triggering the swallowing reflex. This takes part in the pharyngeal phase of swallowing. By rephrasing, this context could be cleared up. In the introductory section it is stated: “Patients with dysphagia, especially post-stroke, show a 54 reduced capability to propel bolus into the pharynx through tongue musculature [7,8].” Here, it should at least be mentioned that part of impairment of the oral phase is not only caused by muscle weakness but also by apraxia. This is relevant particularly with regard to patients suffering from post-stroke dysphagia and will likely not be improved by a device like ToTo that aims at muscle training. With regard to training of muscles and swallowing safety, EMST (expiratory muscle strength training) has been applied in PD patients. As this device does not aim at strengthening the tongue, it may be appealing to combine training devices. This may be an aspect that could be added to the discussion. Line 190: a reference was not found – please add the respective reference. In the methodology section of the article, subjects that were enrolled were classified as being without dysphagic problems. How was this assessed? If it was assessed subjectively, please add this information. Furthermore, was any of the participants taking any medication that may influence your findings? In line 143: the respective number of the figure that is referred to should be added.

Author Response

Please see the attached PDF file.

Reviewer 2 Report

1. The force measurement is based on angular displacement. Dysphagic patients may not be able to exert enough force to move the tongue depressor. What is the minimum force that can be registered in the standard configuration?  The maximum force that can be registered will be limited by range of tongue movement – what are the maximum forces that can be registered for the range of movement of a healthy average tongue?  The authors mention changing the spring constant to adjust these measurable force ranges to the patient’s needs – does this require changing out the springs? How does the flexibility of the tongue depressor affect measurement accuracy?

2. How are non-tongue movements, such as head/neck, prevented from contributing to measured forces?

3. The authors point out that this device is “less invasive” than previous devices because exercise/force measurements occur outside of the mouth, but this may not be an advantage.  There is a question of exercise specificity – for exercises to improve swallowing function, is it enough to strengthen the muscles in any way, or does the training movement have to be similar/identical to the functional behavior that is to be improved?  If so, intraoral devices may have greater specificity to swallowing/bolus formation movements. This topic and its implications for this device should be discussed.

4. Several other devices are mentioned, but I am curious as the perceived advantages/disadvantages of this device vs the Abilex.

5. I agree with the authors that adding instrumentation to automate measurement and track usage would be critical to the usefulness of this device.

6. This is an important topic and an area where improved devices have the potential to improve clinical care, however I am not confident that this device achieves that goal. Future studies on effectiveness to improve not just estimated tongue force (angular displacement), but swallowing function, in a patient population are critical. Such a study should also seek to quantify the patients experience with the device – ease of use, adherence, etc.

Author Response

Please see the attached PDF file.

Round 2

Reviewer 1 Report

Thank you for the revised manuscripts and comments on my considerations. After clearing up the proof-of-principle design this may be added to the heading. The considerations regarding ethics have been cleared up – I was not aware of the discussion that had already happened with the editors. Other than that, all major and minor concerns have been addressed sufficiently.